# Thioacetamide-Induced Acute Liver Injury Increases Metformin Plasma Exposure by Downregulating Renal OCT2 and MATE1 Expression and Function

**DOI:** 10.3390/biomedicines11123314

**Published:** 2023-12-15

**Authors:** Hao Zhi, Yidong Dai, Lin Su, Lu Yang, Wenhan Wu, Zehua Wang, Xinyue Zhu, Li Liu, Jiye Aa, Hanyu Yang

**Affiliations:** 1Center of Drug Metabolism and Pharmacokinetics, School of Pharmacy, China Pharmaceutical University, Nanjing 210009, China; zh_cpu1012@126.com (H.Z.); daiyidong369@163.com (Y.D.); sl201121mm@163.com (L.S.); yanglu2214@163.com (L.Y.); njwuwenhan@126.com (W.W.); zetawilliam813@gmail.com (Z.W.); zxy0191@stu.cpu.edu.cn (X.Z.); liulee@cpu.edu.cn (L.L.); 2Jiangsu Provincial Key Laboratory of Drug Metabolism and Pharmacokinetics, State Key Laboratory of Natural Medicines, China Pharmaceutical University, Nanjing 210009, China

**Keywords:** liver failure, estrogen, tumor necrosis factor-α, organic cation transporter, multidrug and toxin extrusion transporters, metformin

## Abstract

Metformin plasma exposure is increased in rats with thioacetamide (TAA)-induced liver failure. The absorption, distribution, and excretion process of metformin is mainly mediated by organic cation transporters (OCTs) and multidrug and toxin extrusion transporters (MATEs). To investigate the mechanisms of the increase in TAA-induced metformin plasma exposure, we employed intestinal perfusion and urinary excretion assays to evaluate the changes in the absorption and excretion of metformin and used Western blotting to investigate the metformin-related transport proteins’ expression changes and mechanisms. The results showed that neither intestinal OCT2 expression nor metformin intestinal absorption were significantly altered by TAA-induced liver failure, while significantly decreased expression and function of renal OCT2 and MATE1 as well as impaired metformin excretion were observed in TAA rats. HK-2 cells were used as an in vitro model to explore the mechanism of liver-failure-mediated downregulation in renal OCT2 and MATE1. The results demonstrated that among numerous abnormal substances that changed in acute liver failure, elevated estrogen levels and tumor necrosis factor-α were the main factors mediating the downregulation of OCT2 and MATE1. In conclusion, this study highlights the downregulation of renal OCT2 and MATE1 in liver injury and its regulatory mechanism and reveals its roles in the increase in TAA-mediated metformin plasma exposure.

## 1. Introduction

Liver injury is a worldwide common disease with a high incidence and various etiologies, which can be divided into acute liver injury (ALI) and chronic liver injury (CLI) according to its courses [1,2]. Attenuated liver function in liver injury is well known. However, growing pieces of evidence show that liver injury not only alters the hepatic metabolism of drugs but also influences the moieties that are absorbed by the intestine or excretion by the kidney [3]. For example, severe liver failure can induce hepatic nephropathy [4,5]. In hepatic nephropathy, the failure of the renal excretion of drugs due to impaired renal function may affect drug safety and efficacy [6]. On the other hand, gastrointestinal function may also be disrupted during the pathological process of liver injury, resulting in changes in drug absorption [7].

Metformin is a widely used drug to treat diabetes, and clinical studies have reported that its plasma exposure is often altered in patients with liver failure [8]. We also have reported that BDL-induced liver failure increases the plasma exposure of metformin independent of hepatic metabolism changes [9]. Moreover, our preliminary results demonstrated that the plasma concentration of metformin was significantly elevated in TAA-induced ALI rats. Normally, metformin is excreted as a prototype and not affected by hepatic metabolism [10], indicating that the absorption or execration of metformin may be altered in rats with liver failure. The alternation of intestinal or renal metformin transporters caused by ALI may be responsible for this change.

Both organic cation transporters (OCTs) and multi-drug and toxin elimination transporters (MATEs) are involved in metformin transport. OCTs belong to the SLC22A family and MATEs belong to the SLC47A family; they are polyspecific transporters exhibiting broadly overlapping substrate selectivity [11]. OCTs consist of OCT1-3, and MATEs are made up of MATE1, MATE2, and MATE2-K [12]. OCTs mediate cationic compound transport, whose substrates include drugs such as metformin, cisplatin, and cimetidine, as well as endogenous substrates, such as thiamine, creatinine, and acetylcholine [13,14]. OCT1 is mainly expressed on the sinusoidal membrane of hepatocytes, while OCT1, OCT2, and OCT3 are all expressed in the small intestine [15]. In the kidney, OCT2 expressed in the basolateral membrane of the renal tubules, together with MATE1 expressed on the lumen side, mediate the excretion of cationic substances [16]. Metformin is a better substrate for renal OCT2 than hepatic OCT1, and renal OCT2 plays a dominant role in metformin pharmacokinetics [17]. Meanwhile, the intestinal absorption of metformin in rats is also mainly mediated by OCT2 [9]. In the kidneys of rats, metformin is mainly taken in through OCT2 expressed on the basement membrane side and effluxed through MATE1 expressed on the luminal membrane side [18,19]. Numerous studies have revealed the functional interplay between the OCT2-mediated uptake and MATE1-mediated efflux of metformin [20,21,22]. The disruption of MATE1 dramatically reduced metformin secretion in mice [23]. Similarly, metformin-induced lactic acidosis was observed in MATE1 knockout mice [19]. In OCT1/OCT2 knockout mice, the liver to plasma concentration ratio and kidney to plasma concentration ratio of metformin were reduced, which were attributed to the lower OCT1 uptake of metformin in the liver and OCT2 secretion of metformin in the kidney, respectively [24].

Several studies have investigated the regulation of OCT2 and MATE1; for example, acute inflammation elicited by a viral infection downregulated the expression of renal drug transporters, including OCT2 and MATE1 [25]. Similarly, Schmidt-Lauber et al. reported that the cytokines TNF-α, IL-6, and IL-1β decreased the expression of MATE1 mRNA and protein in MATE1-transfected HEK-293 cells [26]. The mRNA expression of OCT2 and MATE1 was found to be decreased in endotoxin-infected mice, which were characterized by elevated serum levels of IL-6, TNF-α, and IFN-γ [27]. In a rat model of hyperuricemia, the mRNA levels of MATE1 and OCT2 were decreased, leading to an increased accumulation of metformin in kidney tissue [28]. The protein expression of MATE1 and OCT2 was reported to be decreased in mice treated with estradiol and the plasma of metformin was significantly elevated [29]. Liver injury can lead to disorders of various endogenous substances, including the production of inflammatory factors, sex hormone metabolism disorders, and reduced ammonia clearance. Whether these disordered endogenous substances are factors that regulate renal MATE1 or OCT2 transport is one of our research focuses.

In this study, we used TAA to establish a rat model of ALI [30,31,32]. Metformin was used as a probe drug to research alterations in the expression and function of OCTs in the intestine and kidney of TAA-induced liver failure. HK-2 cells were used as an in vitro model to further investigate which endogenous substance was capable of regulating OCTs or MATEs [33]. The purpose of the study is to (1) investigate the changes in organic cation transporters in the kidney and intestine during ALI; and (2) elucidate the potential regulatory mechanisms of ALI in downregulating OCT2 and MATE1.

## 2. Materials and Methods

### 2.1. Reagents

Thioacetamide was acquired from J&K Scientific Ltd. (Beijing, China). Kits for activity assays of alanine aminotransferase (ALT), aspartate aminotransferase (AST), content detection of ammonia, and total bilirubin were obtained from Jiancheng Bioengineering Institute (Nanjing, China). Newborn bovine serum, fetal bovine serum, Dulbecco’s modified Eagle medium nutrient mixture F-12, and Dulbecco’s modified Eagle medium were acquired from Invitrogen (Carlsbad, CA, USA). Monoclonal antibodies for OCT2 and MATE1 were obtained from Abcam (Cambridge, UK) and ABclonal (Wuhan, China). Antibody for β-actin was acquired from Bioworld (Louis Park, MN, USA). Human TNF-α was obtained from MCE (MedChemExpress, Monmouth Junction, NJ, USA). All other chemicals were commercially available.

### 2.2. Animals

Male Sprague–Dawley (SD) rats, weighing 220–250 g, were purchased from Sin-British Sippr/BK Laboratory Animal Ltd. (Shanghai, China). All rats were kept at a constant temperature (22 ± 3 °C) and constant humidity (50 ± 10%), had a 12 h day and night environment, and were given food and water freely. Animal feed was purchased from Jiangsu Synergy Medicine Bioengineering Ltd. (Nanjing, China). The animal experiments were carried out by the guidelines on the Care and Use of Animals developed by the National Advisory Committee for Laboratory Animal Research following ARRIVE guidelines and were approved by the Ethics Committee of the Animal Care Council of the China Pharmaceutical University.

### 2.3. Development of ALI Rats

The rats were randomly divided into ALI and CON groups. TAA-induced ALI rats were developed according to our previous and other reports, in which rats were given 300 mg/kg of TAA for two consecutive days [30,34,35]. In brief, the TAA was dissolved in 0.9% normal saline with a concentration of 100 mg/mL. The ALI rats were intraperitoneally injected with 0.3 mL/100 g of TAA (300 mg/kg) for 2 consecutive days, with an interval of 24 h each time, and the rats in the CON group received an equal volume of normal saline. All rats were given a normal diet. To prevent hypoglycemia and electrolyte imbalance, which may occur after modeling, 3% glucose, 0.3% sodium chloride, and 0.149% potassium chloride were added to the drinking water of rats. Twenty-four hours after the second administration, the animals were prepared for the following experiments. 

### 2.4. Pharmacokinetics of Metformin after Oral or Intravenous Administration to Rats

Metformin was dissolved in the normal saline with a concentration of 5 mg/mL. The overnight-fasted rats were orally (40 mg/kg) or intravenously (10 mg/kg) given metformin solution. Specifically, the rats were given 0.8 mL/100 g metformin orally and 0.2 mL/100 g intravenously, respectively. The dosage of metformin was cited based on previous reports [36]. Blood samples (about 200 μL) were collected into heparinized microcentrifuge tubes via the oculi chorioidea vein at designated times (15, 30, 45, 60, 90, 120, 240, and 360 min for oral dose; 5, 10, 15, 30, 60, 90, 120, 240, and 360 min for intravenous dose). Plasma concentrations of metformin were measured by the HPLC method [37]. Briefly, 100 uL blood samples were added into 1.5 mL tubes containing 200 uL acetonitrile. The supernatant was obtained via centrifugation at 16,000× *g* for 10 min. The levels of metformin were determined by a UV detector (235 nm). Separation of metformin was performed on the C18 column (Inertsil ODS-SP, 4.6 × 150 nm, 5 μm). The mobile phase consisted of a mixture of phosphate buffer (2 mmol/L SDS, 0.1% triethylamine, and 0.08% phosphoric acid) and acetonitrile (35:65, *v*/*v*). Flow rate and column temperature were 1.0 mL/min and 40 °C.

### 2.5. Intestinal Absorption of Metformin in Rat Duodenum

Absorption of metformin in rat duodenum was evaluated by in situ single-pass perfusion as in a previous protocol [38]. To put it briefly, the duodenum segment of rats was separated by about 10 cm, and two cannulas were inserted into each end of the segment. A constant-flow pump was connected to the upper cannula for perfusion. The isolated intestinal segment was perfused with preheated Krebs–Hensley buffer (37 °C) for 10 min, followed by Krebs–Hensley buffer containing metformin (50 μg/mL) at 0.2 mL/min for 30 min. The outflow was collected into a distal cannula every 15 min for 90 min. The concentration of metformin in outflow was measured by HPLC. At the end of perfusion, the rats were sacrificed, and the area of perfused intestinal segment A (cm^2^) was measured. The apparent permeability coefficient (P_eff_) was calculated by the following formula:P_eff_ = −Q × ln (C_out_/C_in_)/A,
where Q was the flow rate and C_in_ was the metformin concentration in the inflow. C_out_ was the metformin concentration in the outflow and was corrected by the weight calibration method. A was the area of the perfused intestinal segment.

### 2.6. Urinary Secretion of Metformin

The rats in each group were placed in a metabolic cage in advance for adaptation. After 12 h adaptation, the rats in each group received 10 mg/kg metformin through the tail vein. Urine was collected from each group at 0–6 h and 0–24 h, and the urine volume of rats was recorded. The metformin concentration in urine was then measured, and the excretion fraction of metformin was calculated by the following formula:F (%) = C_urine_ × V_urine_/Dose,
where C_urine_ was the concentration of metformin in urine and V_urine_ was urine volume.

### 2.7. Western Blot 

The animals were sacrificed by rat spinal cord dislocation method under isoflurane anesthesia and the tissues were immediately collected. The protein levels of OCT2 and MATE1 in cells or tissues (intestine, kidney) of rats were measured by Western blotting, as in the previous method [39]. Briefly, samples were homogenized by RIPA lysis buffer containing 1 mmol/L phenyl-methyl-sulfonylfluoride. Protein concentrations were determined via the BCA kit. Equal amounts of proteins were electrophoresed on sodium dodecyl sulfate-polyacrylamide gel (10%) and transferred to nitrocellulose membranes (Millipore, Billerica, MA, USA). Protein was blocked with 5% nonfat dry milk solution dissolved in Tris-buffered saline containing 0.1% Tween 20 (TBST) at room temperature for 1.5 h and then washed with TBST. The membranes were incubated overnight with primary rat monoclonal anti-MATE1 (1:1000 dilution), anti-OCT2 (1:3000 dilution), and anti-β-ACTIN (1:10,000 dilution) at 4 °C. The membranes were washed with TBST three times and then incubated with the secondary anti-rabbit antibody (1:3000 dilution) or anti-rat antibody (1:3000 dilution) for 1.5 h. The protein levels were detected via Tanon high-sig ECL Western blotting substrate (Thermo Fisher Scientific, Waltham, MA, USA) using a gel imaging system (Tanon 5200 Multi Chemiluminescent System, Shanghai, China). All protein levels were normalized to β-actin. 

### 2.8. Real-Time RT-PCR Analysis mRNA Levels of OCT2 and MATE1 in the Kidney of Rats

The mRNA levels of OCT2 and MATE1 in the kidneys were measured using qRT-PCR testing [40]. Total RNAs were extracted from rat kidneys using Trizol and cDNA was synthesized by using the cDNA Reverse Transcription Kit (Novezyme). PCR primer sequences are shown in Appendix A. qRT-PCR was performed on an ABI7500 Fast RT-PCR System (Thermo Fisher, Waltham, MA, USA) for relative quantification. Relative mRNA expression levels were normalized by β-actin expressions (2 ^−∆∆Ct^).

### 2.9. Cell Culture of HK-2 and Drug Treatment

HK-2 cells were provided by JENNIO Biological Technology Ltd. (Guangzhou, China). HK-2 cells were seeded in 12-well plates and cultured with DMEM/F-12 medium supplemented with 10% fetal bovine serum in a humidified incubator with 5% CO_2_ and 95% O_2_ at 37 °C. The mediums were changed every 2 days. When cells reached 80% confluence, the cells were used for the following experiments. Following 48 h incubation with medium containing 20% rat serum, the cells were collected for measuring protein expressions of OCT2 and MATE1. Concentration-dependent effects of IL-6 (500 ng/L, 1 μg/L and 2 μg/L), TNF-α (500 ng/L, 1 μg/L and 2 μg/L), NH_4_Cl (200 μmol/L, 1 mmol/L, and 5 mmol/L), and estrogen (100 ng/L, 300 ng/L and 1 μg/L) on OCT2 and MATE1 levels were measured.

### 2.10. Statistical Analysis

All data are presented as mean ± standard derivation (SD). *t*-tests were used for assessing the two groups and a one-way ANOVA followed by the least significant difference test were used to assess the comparisons among multiple groups. *p* < 0.05 was regarded as indicating statistical significance.

## 3. Results

### 3.1. Biochemical and Physiological Parameters of ALI Rats

TAA-induced ALI was confirmed by the evaluation of the rats’ physiological and biochemical parameters (Table 1). As expected, the body weight, liver weight, and spleen weight of the ALI rats were significantly lower than those of the control rats, and the levels of alanine aminotransferase (ALT), aspartate aminotransferase (AST), total bilirubin (TBL), and serum ammonia of the ALI rats were significantly higher than those of the control rats, indicating that the TAA-induced liver failure model was successfully developed.

### 3.2. Plasma Concentration of Metformin in Rats after Metformin Gavage

The plasma concentrations of metformin in the ALI and control rats were measured after an oral dose of 40 mg/kg metformin. The results showed that TAA-induced ALI significantly increased the plasma concentration of metformin in the rats (Figure 1A). The pharmacokinetic parameters like the peak concentration (C_max_) and the area under curve (AUC) were significantly increased (Figure 1B,C) in the ALI rats, while the clearance (CL) of metformin was significantly decreased, and the mean retention time (MRT) of metformin was significantly prolonged (Figure 1D,E). 

### 3.3. Alternations in Intestinal Absorption of Metformin in ALI Rats

We first tried to explain the alternation in TAA-induced metformin plasma exposure from the perspective of absorption. An in situ single-pass perfusion of the intestine was performed to evaluate the absorption of metformin (Figure 2A,B). Compared with the control rats, the value of P_eff_ and the cumulative absorption had no significant changes in the TAA-induced ALI rats, which indicated that the intestinal absorption of metformin was not altered. The absorption of metformin is mainly mediated by intestinal OCT2. The protein expression of OCT2 on different intestinal segments was measured. As the results showed, there were few differences in OCT2 protein levels in these three intestinal segments between the ALI and CON rats (Figure 2C–E), proving that TAA-induced ALI had no significant effect on the intestinal absorption of metformin.

### 3.4. Plasma Concentration of Metformin in Rats after Intravenous Injection

To verify whether urinary excretion affected the changes in metformin’s pharmacokinetic behavior, we injected 10 mg/kg metformin via the rats’ tail vein and measured the plasma concentration of metformin. Similar to the results of the gavage, the plasma concentrations of metformin in the ALI rats were significantly increased after the intravenous injection of metformin (Figure 3A), and pharmacokinetic parameters such as the C_max_ (Figure 3B) and the AUC (Figure 3C) were significantly increased while the CL (Figure 3D) was decreased, suggesting that decreased metformin excretion may be the main reason for the elevated plasma concentration of metformin in the ALI rats. 

### 3.5. ALI Downregulates Renal OCT2 and MATE1 Expression to Decrease Metformin Excretion

As metformin is mainly excreted by the kidney, urinary excretion experiments were performed for the ALI rats. The urine from the ALI and CON rats was collected 0–6 h and 0–24 h after the intravenous injection of metformin (10 mg/kg), and the concentrations of metformin in the urine were determined respectively (Figure 4A,B). As the results depict, there were no significant differences in the excretion of metformin in CON rats at 0–6 h and 0–24 h, suggesting that, under normal conditions, the metformin was almost excreted within 6 h (Figure 4A,B). More importantly, the cumulative excretion fraction of metformin at 0–6 h and 0–24 h in the ALI rats was significantly lower than that in the control rats, suggesting that TAA-induced ALI significantly reduced the renal excretion of metformin in the rats. The renal excretion of metformin is mainly mediated by renal OCT2 and MATE1 expression. We found that both renal OCT2 and MATE1 protein and the mRNA levels were significantly decreased in the ALI rats (Figure 4C–E), which explained the reduced metformin excretion. 

In addition, the plasma concentrations of creatinine and urea nitrogen were measured to evaluate the renal function of the ALI rats. Interestingly, ALI significantly elevated the plasma concentrations of creatinine, but did not alter the plasma concentrations of urea nitrogen (Figure 4F).

### 3.6. Mechanisms of ALI Downregulation of Renal OCT2 and MATE1 Expression

We used HK-2 cells as an in vitro model to explore the mechanism of AKI-mediated renal OCT2 and MATE1 downregulation. The data from Western blotting showed that 48 h of incubation with the medium containing 20% serum from the ALI rats significantly reduced the protein levels of OCT2 and MATE1 in HK-2 cells (Figure 5A), which was consistent with the in vivo data showing that ALI downregulated OCT2 and MATE1 expression in the kidney. 

ALI is accompanied by changes in the concentrations of ammonia, TNF-α, and estrogen in ALI rat serum (Table 1). Next, we further investigated which abnormal substances that were changed in the serum of the ALI rats led to the downregulation of OCT2 and MATE1 expression. It turned out that ammonia (Figure 5A) and IL-6 (Figure 5B) had no significant effect on the protein expression of OCT2 and MATE1 in the HK-2 cells. Nevertheless, estrogen showed a concentration-dependent inhibition of the expression of OCT2 (Figure 5D), and TNF-α (Figure 5E) showed a concentration-dependent inhibition of the expression of MATE1. These findings suggested that the elevated TNF-α and estrogen levels in the serum of the ALI rats could decrease the expression of renal MATE1 and OCT2, thereby affecting the renal excretion of metformin in the ALI rats.

## 4. Discussion

Since approved by the FDA, metformin has become the recommended initial treatment for diabetes mellitus type 2 (T2DM). However, the FDA issued a warning that recommended caution in patients with chronic liver disease using metformin [41]. We also have reported that BDL-induced liver failure decreased the plasma exposure of metformin independent of hepatic metabolism changes [9]. However, until now, few studies have revealed the influence of ALI on the disposition of metformin. 

In the present study, we found an elevated level of plasma exposure of metformin in TAA-induced ALI rats after an oral dose of metformin. Since metformin is not metabolized in the liver, the plasma concentration of metformin is mainly influenced by absorption in the gastrointestinal tract and excretion through the kidneys [42]. Data from in vivo intestinal perfusion experiments demonstrated that the intestinal absorption of metformin showed an increased tendency in ALI rats, but was not significantly different from that in CON rats. Meanwhile, the protein levels of OCT2 remained unchanged in the duodenum, ileum, and jejunum of the ALI rats when compared with those of the CON rats. Moreover, elevated plasma metformin exposure was also found in the TAA rats after the intravenous injection of metformin. These results exclude the influence of intestine absorption on changes in metformin plasma exposure by ALI. 

Drugs that are mainly eliminated by hepatic metabolism are not the most common drugs that induce adverse drug reactions or cause severe complications in liver injury [43]. Importantly, severe liver failure can lead to kidney damage or renal excretion function impairment, and thus the excretion of drugs that are excreted by the kidney in a prototype may be impaired [44,45]. For instance, the clearance of cimetidine after multiple oral and intravenous administrations was reduced in patients with liver cirrhosis [46]. Changes in renal function also have significant impacts on the in vivo disposal of metformin. The FDA has cautioned against using metformin in the setting of chronic kidney disease, which impairs the excretion of metformin [8]. These results indicated that the renal excretion of metformin was impaired in ALI, which may be the reason for the elevated plasma metformin exposure. To prove the above deduction, urine samples were collected at two time intervals: 0–6 h and 0–24 h after the intravenous administration of metformin. The results showed that the cumulative excretion fraction of metformin at 6 h and 24 h in the ALI rats was significantly lower than that in the control rats. The TAA-induced ALI was not accompanied by obvious alternations in glomerular filtration, as shown by the result that the urea nitrogen levels were not elevated by ALI, whereas we found that the plasma creatinine levels in the TAA rats were significantly increased compared to those in the CON rats. In contrast to the urea nitrogen levels, creatinine was excreted in the urine by OCT2, and the elevated plasma concentrations of creatinine may be attributed to the decreased expression of OCT2 in the ALI rats. In line with these findings, both the protein and mRNA levels of OCT2 and MATE1 were found to be decreased in the renal systems of the TAA rats, suggesting that TAA-induced liver failure decreases renal OCT2 and MATE1 expression to influence the excretion of metformin.

OCT2 and MATE have been reported to be predominantly expressed in renal proximal tubular epithelial cells [33,47]. HK-2 cells are an immortalized proximal tubule epithelial cell line from normal adult human kidneys and retain the functional characteristics of the proximal tubular epithelium [48]. HK-2 cells have been extensively used to study the functions of renal OCT2 and MATE1 [49,50]. To explore the mechanism underlying the reduction in renal MATE1 and OCT2 expression in TAA-induced ALI, HK-2 cells were used as an in vitro model. After culturing the cells with 20% ALI rat serum, the protein levels of MATE1 and OCT2 were significantly decreased, which indicates that altered endogenous compounds in plasma may be the factors that regulate MATE1 and OCT2. Plasma concentrations of ammonia, TNF-α, IL-6, and estrogen were reported to be elevated in ALI rats [51,52,53,54,55,56]. Subsequently, we cultured HK-2 cells with these abnormally changed substances. The results demonstrate that TNF-α dose-dependently reduces the renal expression of MATE1 while estrogen dose-dependently reduces the renal expression of OCT2, which is in line with reported research [26,29]. These findings imply that TNF-α and estrogen may play crucial roles in decreasing the expression of MATE1 and OCT2, respectively. 

In summary, we investigated the pharmacokinetics of metformin in rats with TAA-induced acute liver injury. Our findings revealed significant changes in the pharmacokinetics of metformin in ALI rats. To further understand the underlying mechanisms, we performed intestinal perfusion and urinary excretion experiments, which allowed us to determine the specific effects of TAA-induced liver injury on metformin absorption and excretion. Interestingly, our results indicated that the altered pharmacokinetics of metformin in TAA rats were primarily attributed to impaired renal excretion rather than changes in absorption. This observation prompted us to explore the expression and function of key transporters involved in metformin excretion in the kidney, specifically OCT2 and MATE1. Through our investigations, we discovered that TAA-induced liver injury negatively affected the renal excretion of metformin by downregulating the expression of the renal transporters OCT2 and MATE1. These transporters play a crucial role in facilitating the excretion of metformin from the kidneys into the urine. The decrease in their expression resulted in the reduced ability of the kidneys to eliminate metformin efficiently. To gain further insight into the underlying mechanisms, we conducted in vitro experiments. Our findings suggested that the abnormally elevated levels of serum TNF-α in TAA rats may be a significant contributing factor to the downregulation of MATE1 expression in the kidneys. Additionally, we observed that the elevated levels of estrogen in TAA rats might be responsible for the decreased expression of OCT2 in the kidneys. TNF-α and E2 are reported as activators of several downstream pathways by activating tumor necrosis factor receptors (TNFRs) and estrogen receptors (ERs) [57,58] and participate in the regulation of downstream signaling pathways, such as MAPK, NF-κB, and JNk [59,60,61,62]. The signaling pathway through which E2 and TNF-α regulate the expression of OCT2 and MATE1 needs to be further studied.

The results of our study provide a comprehensive explanation for the observed increase in the plasma concentration of metformin in TAA rats after both its oral administration and its injection intravenously. The impaired renal excretion caused by TAA-induced liver injury, in combination with the altered expression of key transporters, contributes to the accumulation of metformin in the systemic circulation. Overall, our research sheds light on the complex interplay between liver injury, renal excretion, and the pharmacokinetics of metformin. These findings have implications for understanding the potential impact of liver dysfunction on the therapeutic efficacy and safety of metformin in clinical practice. Further studies in this area could help optimize drug dosing and improve patient care.

## Figures and Tables

**Figure 1 biomedicines-11-03314-f001:**
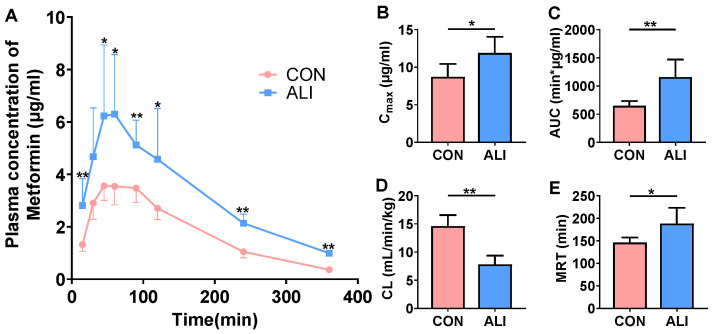
Alternations in plasma concentrations of metformin following oral administration of metformin (40 mg/kg). (**A**) Plasma concentration–time curves of metformin after gavage in TAA-induced ALI rats and CON rats (*n* = 6). (**B**–**E**) Differences in pharmacokinetic parameters of ALI and CON rats. Data are presented as mean ± SD. ** *p* < 0.01 vs. CON. * *p* < 0.05 vs. CON. AUC: area under the curve; CL: clearance; MRT: mean retention time; C_max_: peak concentration; CON: control; ALI: acute liver injury.

**Figure 2 biomedicines-11-03314-f002:**
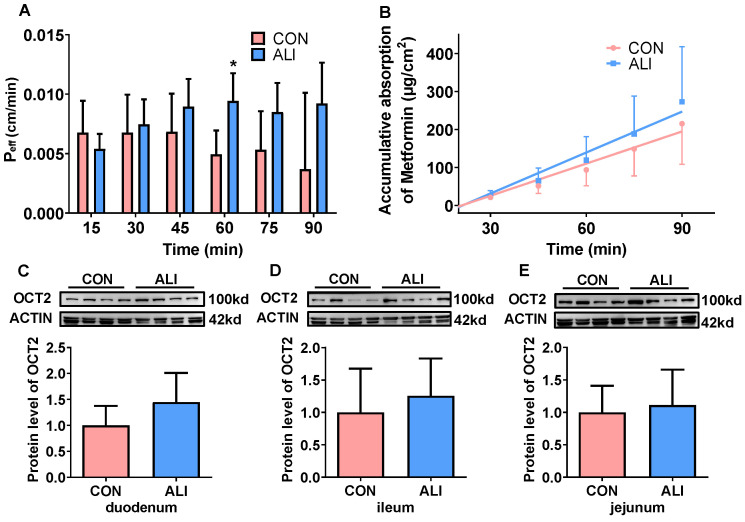
Effects of ALI on the function of OCT2 in the intestine of rats. (**A**) P_eff_ values of metformin in the intestine of rats (*n* = 5). (**B**) Accumulative absorption of metformin in rats (*n* = 5). (**C**–**E**) Protein levels of OCT2 in the duodenum, ileum, and jejunum (*n* = 8). Data are presented as mean ± SD. * *p* < 0.05 vs. CON. OCT2: organic cation transporter 2; CON: control; ALI: acute liver injury.

**Figure 3 biomedicines-11-03314-f003:**
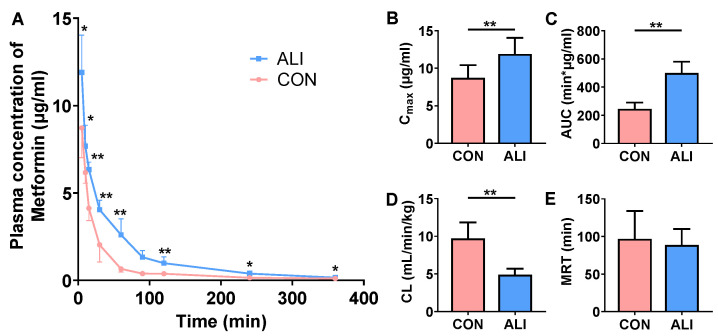
Alternations of plasma concentrations of metformin following intravenous administration of metformin. (**A**) Metformin plasma concentration–time curves after intravenous dose of 10 mg/kg metformin in ALI rats and CON rats. (**B**–**E**) Differences in pharmacokinetic parameters of intravenous administration of metformin in ALI and CON rats. Data are presented as mean ± SD, *n* = 6. ** *p* < 0.01 vs. CON. * *p* < 0.05 vs. CON. AUC: area under the curve; CL: clearance; MRT: mean retention time; C_max_: peak concentration; CON: control; ALI: acute liver injury.

**Figure 4 biomedicines-11-03314-f004:**
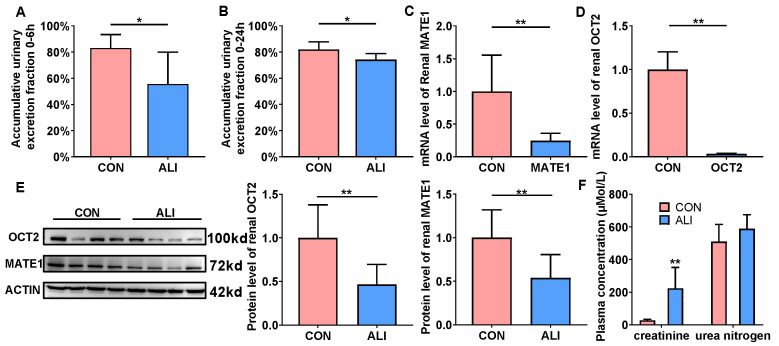
ALI downregulates renal OCT2 and MATE1 expression to decrease metformin excretion. Accumulative urinary excretion of metformin in ALI and CON rats 0–6 h (**A**) and 0–24 h (**B**) (*n* = 6). mRNA level of MATE1 (**C**) and OCT2 (**D**) in kidneys of ALI and CON rats (*n* = 6). (**E**) Protein levels of OCT2 and MATE1 in the kidneys of ALI and CON rats (*n* = 8). (**F**) Plasma concentrations of creatinine and urea nitrogen for CON and ALI rats (*n* = 6). Data are presented as mean ± SD. ** *p* < 0.01 vs. CON. * *p* < 0.05 vs. CON. OCT2: organic cation transporter 2; MATE1: multi-drug and toxin extrusion transporters; CON: control; ALI: acute liver injury.

**Figure 5 biomedicines-11-03314-f005:**
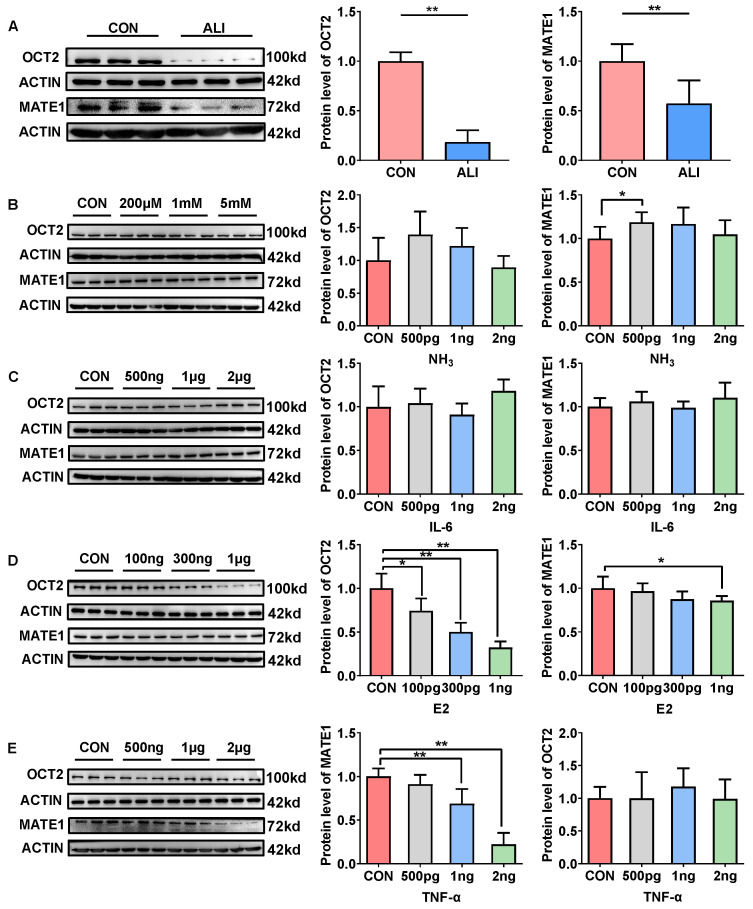
Mechanism of ALI downregulation of renal OCT2 and MATE1 expression. Effects of serum of ALI and CON rats on the protein expression of OCT2 and MATE1 in HK-2 cells (**A**). Effects of dose-elevated ammonia (**B**), IL-6 (**C**), E2 (**D**), and TNF-a (**E**) on the protein levels of MATE1 and OCT2 in HK-2 cells. Data are presented as mean ± SD, *n* = 6. ** *p* < 0.01 vs. CON. * *p* < 0.05 vs. CON. OCT2: organic cation transporter 2; MATE1: multi-drug and toxin extrusion transporters; CON: control; ALI: acute liver injury.

**Table 1 biomedicines-11-03314-t001:** Physiological and biochemical parameters of CON and ALI rats.

Parameters	CON	ALI
Body weight (BW) (g)	236.71 ± 15.85	216.40 ± 6.29 **
Liver weight (% BW)	2.75 ± 0.17	4.18 ± 0.24 **
Spleen weight (% BW)	0.25 ± 0.04	0.22 ± 0.03
ALT (IU/L)	20.35 ± 13.51	88.08 ± 38.28 **
AST (IU/L)	10.21 ± 6.90	39.72 ± 2.65 **
TBL (μmol/L)	ND	3.61 ± 1.19 **
Serum ammonia (μmol/L)	182.32 ± 13.26	322.75 ± 46.26 **
Estrogen (pg/mL)	27.06 ± 2.93	86.42 ± 38.42 **
TNF-α (pg/mL)	5.82 ± 2.17	165.23 ± 32.00 **

Data are presented as mean ± SD (*n* = 6). ** *p* < 0.01 vs. CON. CON: control; ALI: acute liver injury; ALT: alanine aminotransferase; AST: aspartate aminotransferase; TBL: total bilirubin; ND: not detected.

## Data Availability

The data presented in this study are all contained within the main body of this article.

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
