# Peer review of "Thioacetamide-Induced Acute Liver Injury Increases Metformin Plasma Exposure by Downregulating Renal OCT2 and MATE1 Expression and Function"

_biomedicines, 2023, doi:10.3390/biomedicines11123314_

Round 1
Reviewer 1 Report
Comments and Suggestions for Authors
Review of the paper entitled „Thioacetamide-induced acute liver injury increases metformin plasma exposure by downregulating renal OCT2 and MATE1 expression and function” by Hao Zhi, Yi-dong Dai, Lin Su, Lu Yang, Wen-han Wu, Ze-hua Wang, Xin-yue Zhu, Li Liu, Ji-ye A and Han-yu Yang
The Authors established rat thioacetamide (TAA)-induced acute liver injury model (ALI). Next the Authors used metformin as a probe drug to research alterations in the expression and function of organic cation transporters (OCTs) and multi-drug and toxin elimination transporters (MATEs) in the intestine and kidney of ALI rats. The obtained results indicated elevated plasma exposure of metformin in ALI rats after both oral and intravenous administration of metformin. The Authors discovered also that TAA-induced liver injury negatively affected the renal excretion of metformin by downregulating the expression of renal transporters OCT2 and MATE1. In studies conducted on the HK-2 cell line, the Authors also demonstrated that the TNF-α and estrogen could inhibit the expression of renal renal transporters OCT2 and MATE1.
This is an interesting paper.
My comments
(1) I have great doubts regarding the dose of TAA. TAA is a very toxic compound and according to available data, the LD 50 of TAA for rats after oral administration is 301 mg/kg. The dose of 300 mg/kg used by the Authors is close to lethal dose. Additionally, the Authors administered TAA intraperitoneally, while the LD50 value of 301 mg/kg is for the oral administration. Typically, the LD 50 values of chemical compounds for intraperitoneal administration are several times lower than the LD50 values for oral administration. Did any animals die during the experiment? The Authors need to explain these problems in their manuscript.
(2) The Authors should provide more details in the Materials and Methods section. In what did the Authors dissolve TAA?
What volume of solution containing TAA and volume of salt were administered intraperitoneally to the animals?
What volume of solution containing metformin and volume of salt were administered intravenous to the animals?
How blood from animals was obtained?
How the animals were killed?
(3) The Authors were used HK-2 cell as an in vitro model to investigate which endogenous substance was capable to regulate OCTs or MATEs. I would like the Authors to explain in their manuscript why they chose the HK-2 cell line for their research.

Author Response
Reviewers' Comments to Author:
REVIEWER 1:
The Authors established rat thioacetamide (TAA)-induced acute liver injury model (ALI). Next the Authors used metformin as a probe drug to research alterations in the expression and function of organic cation transporters (OCTs) and multi-drug and toxin elimination transporters (MATEs) in the intestine and kidney of ALI rats. The obtained results indicated elevated plasma exposure of metformin in ALI rats after both oral and intravenous administration of metformin. The Authors discovered also that TAA-induced liver injury negatively affected the renal excretion of metformin by downregulating the expression of renal transporters OCT2 and MATE1. In studies conducted on the HK-2 cell line, the Authors also demonstrated that the TNF-α and estrogen could inhibit the expression of renal transporters OCT2 and MATE1.
This is an interesting paper.
Point 1:
I have great doubts regarding the dose of TAA. TAA is a very toxic compound and according to available data, the LD 50 of TAA for rats after oral administration is 301 mg/kg. The dose of 300 mg/kg used by the Authors is close to lethal dose. Additionally, the Authors administered TAA intraperitoneally, while the LD50 value of 301 mg/kg is for the oral administration. Typically, the LD 50 values of chemical compounds for intraperitoneal administration are several times lower than the LD50 values for oral administration. Did any animals die during the experiment? The Authors need to explain these problems in their manuscript.
Response 1:
Thanks for your suggestions. We are sorry not to illustrate it clearly. The dosage of TAA was cited based on previous reports [1, 2], in which rats were given 300 mg/kg of TAA for two consecutive days to induce acute liver failure. TAA dose has also been validated in our previous studies [3-5]. Hypoglycemia and electrolyte imbalance are related to the morality of rats after modeling. To avoid this, 3% glucose, 0.3% sodium chloride, and 0.149% potassium chloride were added to the drinking water. With this method, the mortality rate of TAA rats was about 10%.
Reference:
[1] A.R. Jayakumar, V. Valdes, M.D. Norenberg, The Na-K-Cl cotransporter in the brain edema of acute liver failure, J Hepatol 54(2) (2011) 272-8.
[2] R. Bruck, H. Aeed, Y. Avni, H. Shirin, Z. Matas, M. Shahmurov, I. Avinoach, G. Zozulya, N. Weizman, A. Hochman, Melatonin inhibits nuclear factor kappa B activation and oxidative stress and protects against thioacetamide induced liver damage in rats, J Hepatol 40(1) (2004) 86-93.
[3] F. Wang, M.X. Miao, B.B. Sun, Z.J. Wang, X.G. Tang, Y. Chen, K.J. Zhao, X.D. Liu, L. Liu, Acute liver failure enhances oral plasma exposure of zidovudine in rats by downregulation of hepatic UGT2B7 and intestinal P-gp, Acta Pharmacol Sin 38(11) (2017) 1554-1565.
[4] S. Jin, X.T. Wang, L. Liu, D. Yao, C. Liu, M. Zhang, H.F. Guo, X.D. Liu, P-glycoprotein and multidrug resistance-associated protein 2 are oppositely altered in brain of rats with thioacetamide-induced acute liver failure, Liver Int 33(2) (2013) 274-82.
[5] Y. Li, J. Zhang, P. Xu, B. Sun, Z. Zhong, C. Liu, Z. Ling, Y. Chen, N. Shu, K. Zhao, L. Liu, X. Liu, Acute liver failure impairs function and expression of breast cancer-resistant protein (BCRP) at rat blood-brain barrier partly via ammonia-ROS-ERK1/2 activation, J Neurochem 138(2) (2016) 282-94.
According to your advice, we rewrote the “Material and Method” section in the revised manuscript as follows:
In “Material and Method”
2.3 Development of ALI rats
The rats were randomly divided into ALI and CON groups. TAA-induced ALI rats were developed according to our previous and other reports, in which rats were given 300 mg/kg of TAA for two consecutive days [30,34,35]. In brief, the TAA was dissolved in 0.9% normal saline with a concentration of 100 mg/mL. The ALI rats were intraperitoneally injected with 0.3 ml/100g of TAA (300 mg/kg) for 2 consecutive days, with an interval of 24 h each time, and the rats in the CON group received an equal volume of normal saline. All rats were given a normal diet. To prevent hypoglycemia and electrolyte imbalance that may occur after modeling, 3% glucose, 0.3% sodium chloride, and 0.149% potassium chloride were added to the drinking water of rats.
Point 2:
The Authors should provide more details in the Materials and Methods section. In what did the Authors dissolve TAA? What volume of solution containing TAA and volume of salt were administered intraperitoneally to the animals? What volume of solution containing metformin and volume of salt were administered intravenous to the animals? How blood from animals was obtained? How the animals were killed?
Response 2:
Thank you for your suggestions. We are sorry not to illustrate it clearly. According to your suggestions, we rewrote the “Materials and Methods” section as follows:
For Point 2a: In what did the Authors dissolve TAA? What volume of solution containing TAA and volume of salt were administered intraperitoneally to the animals?
In “Material and Method”:
2.3 Development of ALI rats
The TAA was dissolved in 0.9% normal saline with a concentration of 100 mg/mL. The ALI rats were intraperitoneally injected with 0.3 ml/100g of TAA (300 mg/kg) for 2 consecutive days, with an interval of 24 h each time, and the rats in the CON group received an equal volume of normal saline.
Point 2b: What volume of solution containing metformin and volume of salt were administered intravenous to the animals? How blood from animals was obtained?
In “Material and Method”:
2.4 Pharmacokinetics of metformin after oral or intravenous administration to rats
Metformin was dissolved in the normal saline with a concentration of 5 mg/mL. The overnight fasted rats were orally (40 mg/kg) or intravenously (10 mg/kg) given metformin solution. Specifically, the rats were given metformin 0.8 mL/100g orally and 0.2 ml/100g intravenously, respectively. The dosage of metformin was cited based on previous reports [38]. Blood samples (about 200 mL) were collected into heparinized microcentrifuge tubes via the oculi chorioidea vein at designated times (15, 30, 45, 60, 90, 120, 240, and 360 min for oral dose; 5, 10, 15, 30, 60, 90, 120, 240, and 360 min for intravenous dose). Plasma concentrations of metformin were measured by the HPLC method [39].
Point 2c: How the animals were killed?
In “Material and Method”:
2.7 Western Blot
The animals were sacrificed by rat spinal cord dislocation method under isoflurane anesthesia and the tissues were immediately collected. The protein levels of OCT2 and MATE1 in cells or tissues (intestine, kidney) of rats were measured by western blot as the previous method [41].
Point 3:
The Authors were used HK-2 cell as an in vitro model to investigate which endogenous substance was capable to regulate OCTs or MATEs. I would like the Authors to explain in their manuscript why they chose the HK-2 cell line for their research.
Response 3:
We are sorry not to clearly illustrate it. HK-2 cells are an immortalized proximal tubule epithelial cell line from normal adult human kidneys [6], and the OCT2 and MATE are predominantly expressed in renal proximal tubular epithelial cells. HK-2 cells have been extensively used to study the functions of renal OCT2 and MATE1 [7-10]. Therefore, we choose HK-2 cells to explore the mechanisms of liver failure-mediated downregulation in OCT2 and MATE1.
References:
[6] M.J. Ryan, G. Johnson, J. Kirk, S.M. Fuerstenberg, R.A. Zager, B. Torok-Storb, HK-2: an immortalized proximal tubule epithelial cell line from normal adult human kidney, Kidney Int 45(1) (1994) 48-57.
[7] S. Yang, Y. Dai, Z. Liu, C. Wang, Q. Meng, X. Huo, H. Sun, X. Ma, J. Peng, K. Liu, Involvement of organic cation transporter 2 in the metformin-associated increased lactate levels caused by contrast-induced nephropathy, Biomed Pharmacother 106 (2018) 1760-1766.
[8] Q. Shen, J. Wang, Z. Yuan, Z. Jiang, T. Shu, D. Xu, J. He, L. Zhang, X. Huang, Key role of organic cation transporter 2 for the nephrotoxicity effect of triptolide in rheumatoid arthritis, Int Immunopharmacol 77 (2019) 105959.
[9] M.X. Wang, Y.L. Liu, Y. Yang, D.M. Zhang, L.D. Kong, Nuciferine restores potassium oxonate-induced hyperuricemia and kidney inflammation in mice, Eur J Pharmacol 747 (2015) 59-70.
[10] S. Hong, S. Li, X. Meng, P. Li, X. Wang, M. Su, X. Liu, L. Liu, Bile duct ligation differently regulates protein expressions of organic cation transporters in intestine, liver and kidney of rats through activation of farnesoid X receptor by cholate and bilirubin, Acta Pharm Sin B 13(1) (2023) 227-245.
According to your suggestions, we rewrote the “Discussion” section in the revised manuscript as follows:
In “Discussion”
The OCT2 and MATE have been reported to be predominantly expressed in renal proximal tubular epithelial cells [33,48]. HK-2 cells are an immortalized proximal tubule epithelial cell line from normal adult human kidneys and retain functional characteristics of proximal tubular epithelium [49]. HK-2 cells have been extensively used to study the functions of renal OCT2 and MATE1 [50,51]. To explore the mechanism underlying the reduction of renal MATE1 and OCT2 expression in TAA-induced ALI, HK-2 cells were used as an in vitro model.

Reviewer 2 Report
Comments and Suggestions for Authors
Manuscript title: Thioacetamide-induced acute liver injury increases metformin plasma exposure by downregulating renal OCT2 and MATE1 expression and function.
In the manuscript, the authors aimed to investigate mechanisms of action of metformin plasma exposure under the effects of thioacetamide-induced acute liver injury in vivo and in vitro. In general, the authors have completed a reasonable study with very informative data on the relationships of the increased metformin plasma exposure and the impaired renal function. The statistical analysis and graphic presentation also have been completed in details. However, the presentation of this study may be strengthened by adding the discussion on the treatment of HK-2 cells by using 20% ALI rat serum not by metformin, which might be metabolized to the others, meaning that the products might be the origins.
Specific comments:
1. Please revise the presentation of compartment of the unit and items, it should be spaced in figures.
2. L. 345-346,
All these results suggesting that ALI may also influence renal execration to change plasma metformin exposure.
Q. Please rephrase the sentence.
3. L. 157,
The metformin concentration in urine was then measured.
Q. Please indicate the analysis method. In addition to HPLC analysis, is there a more precise method for the analysis of metabolites of metformin?
Comments on the Quality of English Language
The minor editing of English language is encouraged.
Author Response
REVIEWER 2:
In the manuscript, the authors aimed to investigate mechanisms of action of metformin plasma exposure under the effects of thioacetamide-induced acute liver injury in vivo and in vitro. In general, the authors have completed a reasonable study with very informative data on the relationships of the increased metformin plasma exposure and the impaired renal function. The statistical analysis and graphic presentation also have been completed in details.
Point 1:
However, the presentation of this study may be strengthened by adding the discussion on the treatment of HK-2 cells by using 20% ALI rat serum not by metformin, which might be metabolized to the others, meaning that the products might be the origins.
Response 1:
The treatment of HK-2 cells by using 20% ALI rat serum was to investigate the regulatory mechanisms of ALI-mediated OCT2 and MATE1 downregulation. Given that metformin is excreted as a prototype and not affected by hepatic metabolism [1], we did not consider its metabolism.
Reference:
[1] G.G. Graham, J. Punt, M. Arora, R.O. Day, M.P. Doogue, J.K. Duong, T.J. Furlong, J.R. Greenfield, L.C. Greenup, C.M. Kirkpatrick, J.E. Ray, P. Timmins, K.M. Williams, Clinical pharmacokinetics of metformin, Clin Pharmacokinet 50(2) (2011) 81-98.
Point 2:
Please revise the presentation of compartment of the unit and items, it should be spaced in figures.
Response 2:
Thanks for pointing out these issues. We have checked all the figures and the revised manuscript was resubmitted.
Point 3:
- 345-346, All these results suggesting that ALI may also influence renal execration to change plasma metformin exposure.
Q. Please rephrase the sentence.
Response 3:
We have rephrased the sentence in the revised manuscript as follows:
“These results indicate that the renal execration of metformin was impaired in ALI, which may be the reason for the elevated plasma metformin exposure.”
Point 4:
- 157, The metformin concentration in urine was then measured.
Q. Please indicate the analysis method. In addition to HPLC analysis, is there a more precise method for the analysis of metabolites of metformin?
Response
We are sorry not to clearly illustrate it. Metformin cannot be metabolized in the human body, which is excreted unchanged in urine [2-4]. Therefore, we did not analyze its metabolites. Besides, we have added the conclude method for the measurement of metformin according to your suggestions. The revised manuscript is as follows:
Reference:
[2] G.G. Graham, J. Punt, M. Arora, R.O. Day, M.P. Doogue, J.K. Duong, T.J. Furlong, J.R. Greenfield, L.C. Greenup, C.M. Kirkpatrick, J.E. Ray, P. Timmins, K.M. Williams, Clinical pharmacokinetics of metformin, Clin Pharmacokinet 50(2) (2011) 81-98.
[3] A.J. Scheen, Clinical pharmacokinetics of metformin, Clin Pharmacokinet 30(5) (1996) 359-71.
[4] L. He, Metformin and Systemic Metabolism, Trends Pharmacol Sci 41(11) (2020) 868-881.
In “Material and Method”
2.4 Pharmacokinetics of metformin after oral or intravenous administration to rats
Briefly, 100 uL blood samples were added into 1.5 mL tubes containing 200 uL acetonitrile. The supernatant was obtained by centrifuging at 16000g for 10 minutes. The levels of metformin were determined by a UV detector (235nm). Separation of metformin was performed on the C18 column (Inertsil ODS-SP, 4.6×150 nm, 5μm). The mobile phase consisted of a mixture of phosphate buffer (2 mmol/L SDS, 0.1% triethylamine, and 0.08% phosphoric acid) and acetonitrile (65:35, v/v). Flow rate and column temperature were 1.0 mL/min and 40 ℃.

Reviewer 3 Report
Comments and Suggestions for Authors
The article entitled “Thioacetamide-induced acute liver injury increases metformin plasma exposure by downregulating renal OCT2 and MATE1 expression and function” is an experimental study about the mechanisms that regulate the renal excretion of drugs after liver injury. The authors used the plasma metformin as a probe drug, experimental rats treated with thioacetamide causing acute liver injury, and cell culture of HK2-cells (intestine, kidney) of rats. The results of this study show that the substances from liver injury that are responsible for increased plasma metformin are TNF-α and estrogen. This effect is mediated by kidney reduced expression of the transporter molecules such as OCT2 and MATE1. The project study and methods are well-written and appropriate. I have no any questions. Therefore, I think that this article is suitable for publication in its current version.
Author Response
Point 3:
The article entitled “Thioacetamide-induced acute liver injury increases metformin plasma exposure by downregulating renal OCT2 and MATE1 expression and function” is an experimental study about the mechanisms that regulate the renal excretion of drugs after liver injury. The authors used the plasma metformin as a probe drug, experimental rats treated with thioacetamide causing acute liver injury, and cell culture of HK2-cells (intestine, kidney) of rats. The results of this study show that the substances from liver injury that are responsible for increased plasma metformin are TNF-α and estrogen. This effect is mediated by kidney reduced expression of the transporter molecules such as OCT2 and MATE1. The project study and methods are well-written and appropriate. I have no any questions. Therefore, I think that this article is suitable for publication in its current version.
Response 3:
Thank you for your nice comments on our article.

Round 2
Reviewer 1 Report
Comments and Suggestions for Authors
The authors have significantly improved their manuscript. In this version, the paper can be accepted for publication
Author Response
Point:
The authors have significantly improved their manuscript. In this version, the paper can be accepted for publication.
Response:
Thank you for your nice comments on our article.